Foveavelia, a new South American genus of Veliinae (Hemiptera: Heteroptera: Veliidae)

Rodrigues Higor D. D. higorddr@gmail.com
http://orcid.org/0000-0002-6692-0323 Moreira Felipe F. F.
Laboratório de Entomologia, Instituto Oswaldo Cruz, Fundação Oswaldo Cruz , Rio de Janeiro, Rio de Janeiro , Brazil
Gillespie Joseph
Electronic publication date: 2024 Mar 21
Publication date: 2024
Volume: 12
Electronic Location ID: e16772
Received 2023 Oct 6; Accepted 2023 Dec 18
Copyright: © 2024 Rodrigues and Moreira
Copyright year: 2024
Copyright holder: Rodrigues and Moreira
License: This is an open access article distributed under the terms of the Creative Commons Attribution License, which permits unrestricted use, distribution, reproduction and adaptation in any medium and for any purpose provided that it is properly attributed. For attribution, the original author(s), title, publication source (PeerJ) and either DOI or URL of the article must be cited.
License URL: https://creativecommons.org/licenses/by/4.0/

Keywords: Aquatic insects, Gerromorpha, Neotropics, New combinations, Taxonomy

Funding: Fundação Carlos Chagas Filho de Amparo à Pesquisa do Estado do Rio de Janeiro FAPERJ E-26/202.437/2019; E-26/202.438/2019 Conselho Nacional de Desenvolvimento Científico e Tecnológico (process 301942/2019-6) Fundação Carlos Chagas Filho de Amparo à Pesquisa do Estado do Rio de Janeiro (processes E-26/201.362/2021 and E-26/200.649/2023) Financial support was provided to HDDR by the Fundação Carlos Chagas Filho de Amparo à Pesquisa do Estado do Rio de Janeiro (FAPERJ E-26/202.437/2019; E-26/202.438/2019). FFFM benefited from grants provided by the Conselho Nacional de Desenvolvimento Científico e Tecnológico (process 301942/2019-6) and the Fundação Carlos Chagas Filho de Amparo à Pesquisa do Estado do Rio de Janeiro (processes E-26/201.362/2021 and E-26/200.649/2023). The funders had no role in study design, data collection and analysis, decision to publish, or preparation of the manuscript.

==============================
Background

Semiaquatic bugs (Hemiptera: Heteroptera: Gerromorpha) are distributed worldwide and play fundamental roles in limnic ecosystems. They are the most successful group of organisms to occupy the air-water interface, are important models to study ecology and evolution, and can be relevant tools in biomonitoring. Veliidae is the second most speciose family of semiaquatic bugs, but its internal classification, including subfamilies and genera, is artificial and based on symplesiomorphies. One of these non-monophyletic entities is Paravelia Breddin, 1898, the largest genus in the subfamily Veliinae.

Results

In an effort to better classify the Veliinae, we describe Foveavelia to hold five South American species previously placed in Paravelia. The new genus is characterized by the following combination of features: unusual coarse cuticular punctures throughout the thorax and abdomen; a pair of small, frosty, pubescent areas formed by a very dense layer of short setae on the anterior lobe of the pronotum; fore tibial grasping comb present only in males; middle tibia with a row of elongate dark-brown trichobothria-like setae on the distal third, decreasing in size distally; macropterous specimens with the apical macula of the forewings elongate and constricted at mid-length, reaching the wing apex; and the male proctiger with a pair of anterodorsal projections. Besides the description, a key to the species of Foveavelia is provided, accompanied by illustrations and a species distribution map.

Introduction

Veliidae (Hemiptera: Heteroptera: Gerromorpha) is a family of small to medium-sized insects that live predominantly on the surface of the water. Some of them occur on stagnant waters, such as lakes and puddles, while others occupy rivers and streams, and a few can be found in terrestrial environments relatively far from the water (Schuh & Slater, 1995). Andersen (1982) established a phylogeny for Veliidae based on morphology and proposed the division into six subfamilies: Haloveliinae, Microveliinae, Ocellloveliinae, Perittopinae, Rhagoveliinae and Veliinae. Subsequently, Damgaard (2008) proposed a phylogeny for Gerromorpha based on morphological and molecular data. He demonstrated that the subfamilies Microveliinae and Haloveliinae were actually closer to Gerridae than to Veliidae, and found low support for other clades of Veliidae, such as the subfamily Veliinae.

More recently, Armisén et al. (2022) obtained a phylogeny of the Gerromorpha based on transcriptomes and corroborated with Damgaard (2008), in which Haloveliinae + Microveliinae formed a clade sister to other Gerridae, leaving Veliidae with only four subfamilies. Within Veliidae, the same authors did not recover the monophyly of Veliinae. Currently, this subfamily includes 11 valid genera, eight of which occur in the Western Hemisphere: Altavelia Polhemus & Moreira, 2019; Callivelia D. Polhemus, 2021; Oiovelia Drake & Maldonado-Capriles, 1952; Paravelia Breddin, 1898; Platyvelia Polhemus & Polhemus, 1993; Steinovelia Polhemus & Polhemus, 1993; Stridulivelia Hungerford, 1929; and Veloidea Gould, 1934. The other three genera that occur in the Palearctic, Afrotropical, and Indo-Malay regions are Angilia Stål, 1865; Angilovelia Andersen, 1981; and Velia Latreille, 1804.

The Neotropical genus Paravelia is the most speciose of the Veliinae, with 51 valid species (Rodrigues & Moreira, 2022; Rodrigues, Moreira & Morales, 2022). The paraphyly of this genus was hypothesized by different authors (e.g., Polhemus & Polhemus, 1993; Rodrigues et al., 2014; Rodrigues & Moreira, 2016; Polhemus, 2021), but without a phylogenetic basis. Armisén et al. (2022) recovered the genus as polyphyletic, but assessing the relationships among its species is still premature, because only a few representatives were included in their analysis.

The study of Veliinae species and their respective type specimens allowed us to identify and define a distinct group of five species within Paravelia that share unique characteristics. Based on it, the new genus Foveavelia is here proposed for this group, based mainly on an important diagnostic and distinct feature: the coarse cuticular punctures found throughout the body. We also present an illustrated taxonomic key and a distribution map for the included species.

Materials and Methods

Museum visits

Pinned specimens were examined at the following public collections: DPIC–Departamento de Parasitologia, Universidade Federal de Minas Gerais, Belo Horizonte, Brazil; INPA–Instituto Nacional de Pesquisas da Amazônia, Manaus, Brazil; MZUSP–Museu de Zoologia, Universidade de São Paulo, São Paulo, Brazil; NMNH–National Museum of Natural History, Smithsonian Institution, Washington D.C., United States; UMC–University of Missouri, Columbia, United States.

Morphological study

All measurements are given in millimeters. Antennomeres and abdominal segments numbers are expressed as Roman numerals. We used standard entomological techniques to examine the morphology of the specimens used in this study. Abdominal segment VIII and genital capsule of the males were removed using forceps and an entomological pin, without the need for clarification. Photographs have been obtained using a Leica DFC420 camera attached to a LeicaM165C binocular microscope, processed with the Leica Application Suite V3.7.0, and stacked using Auto-Montage. Scanning electron microscopy photographs have been provided by Dr. Silvia Mazzucconi. All final figures have been prepared using Adobe Photoshop CS6.

Nomenclatural acts

The electronic version of this article in Portable Document Format (PDF) will represent a published work according to the International Commission on Zoological Nomenclature (ICZN), and hence the new names contained in the electronic version are effectively published under that Code from the electronic edition alone. This published work and the nomenclatural acts it contains have been registered in ZooBank, the online registration system for the ICZN. The ZooBank LSIDs (Life Science Identifiers) can be resolved and the associated information viewed through any standard web browser by appending the LSID to the prefix http://zoobank.org/. The LSID for this publication is: (urn:lsid:zoobank.org:pub:DAAB68B0-7AB5-4D92-AAE8-A9051CD9EC11). The online version of this work is archived and available from the following digital repositories: PeerJ, PubMed Central SCIE and CLOCKSS.

Results

Foveavelia Rodrigues & Moreira new genus

(Figs. 1–7)

Figure 1 Foveavelia species, dorsal and ventral views.

(A) Foveavelia amapaensis (Rodrigues et al., 2014), dorsal habitus of male holotype (specimen was destroyed in a fire). (B–D) Foveavelia bilobata (Rodrigues et al., 2014), (B) dorsal habitus of male from Colombia, (C) dorsal, and (D) ventral habitus of male paratype from Brazil, dashed line indicates length of grasping comb. Am, apical macula; bm, basal macula; fb, frosty pubescence.

Figure 2 Structures of Foveavelia anta (Mazzucconi, 2000).

Scanning electron microscopy. (A) Dorsal view of head, pronotum and part of abdomen (wings removed). (B) Pronotal punctures. (C) Male abdomen in dorsal view (wings and genital capsule removed). (D) Part of abdominal mediotergite II with suboval punctures. (E) Suboval puncture of abdominal mediotergite IV in detail. (F) Male abdomen in lateral view (wings and genital capsule removed), white arrow indicates posteroventral margin of abdominal segment VIII acuminating distally. Mt, mediotergite; lt, laterotergite. Photographs provided by Dr. Silvia Mazzucconi.

Figure 3 Foveavelia species, male structures.

(A and B) Ventral view of abdominal apex, (A) F. amapaensis (Rodrigues et al., 2014), (B) F. bilobata (Rodrigues et al., 2014). (C and D) Abdominal segment VIII of F. bilobata, (C) lateral, and (D) ventral views. (E–H) Genital capsule in lateral view, showing proctiger projections detailed in frontal view, (E) F. amapaensis, (F) F. anta (Mazzucconi, 2000), (G) F. bilobata, (H) F. dilatata (Polhemus & Polhemus, 1984).

Figure 4 Foveavelia dilatata (Polhemus & Polhemus, 1984), dorsal views of female and male.

(A and B) Dorsal habitus of brachypterous females, (A) paratype from Brazil, (B) specimen from Brazil. (C) Dorsal view of abdomen of brachypterous female, specimen from Brazil. (D) Dorsal habitus of brachypterous male, paratype from Brazil. (E) Dorsal habitus of macropterous male, specimen from Brazil.

Figure 5 Foveavelia dilatata (Polhemus & Polhemus, 1984), brachypterous female from Peru.

(A) Dorsal, (B) ventral, and (C) lateral habitus.

Figure 6 Male structures of Foveavelia dilatata (Polhemus & Polhemus, 1984), paratype from Brazil.

(A–C) Abdominal segment VIII, (A) lateral, (B) ventral, and (C) dorsal views. (D) Genital capsule in lateral view. (E) Left paramere in lateral view. (F–H) proctiger, (F) dorsal, (G) lateral, and (H) frontal views, white arrow indicates pair of anterodorsal projections.

Figure 7 Foveavelia species, dorsal and lateral views.

(A and B) Foveavelia hungerfordi, macropterous female paratype from Brazil, (A) dorsal, and (B) lateral views. (C–D) Foveavelia sp., macropterous female from Brazil, (C) dorsal, and (D) lateral views. (E) Foveavelia amapaensis, macropterous male holotype in lateral view (specimen was destroyed in a fire). if = impressed furrow.

Type species. Velia hungerfordi Drake & Harris, 1933, by present designation.

Diagnosis. Body length 4.80–6.50; body vestiture composed of moderately dense, erect, thin, brown setae (Figs. 1, 4, 5, 7); anterior lobe of pronotum with a pair of small, frosty, pubescent areas formed by a very dense layer of short setae (sometimes indistinct, as in Fig. 1A); macropterous specimens with apical macula of forewings elongate and constricted at mid-length (Figs. 1A–1C, 4E, 7A, 7C); meso- and metasterna each with a pair of tubercles, these tubercles meeting at suture, forming a cavity between them; thorax, abdominal mediotergites and sterna (except sternum VII) with numerous suboval, coarse cuticular punctures (Fig. 2); male proctiger with a pair of anterodorsal projections (Figs. 3E–3H, 6H).

Description

Medium-sized veliids; macropterous or brachypterous (micropterous and apterous forms unknown); ground color brown, covered by golden pubescence, with faint yellowish annulations on femora. Body moderately robust, length 4.80–6.50 mm; general characteristics and size not sexually dimorphic (Figs. 1, 7), except for F. dilatata (Figs. 4–5).

Head: Declivant anteriorly, not recessed into pronotum, with usual three pairs of trichobothria and impressed median line; posterodorsal region with a pair of narrow, posteriorly convergent, impressed lines located between impressed median line and elliptical indentation on each side (Fig. 2A); gula and buccula with several suboval, coarse punctures; gular region visible (Figs. 7B, 7D–7E). Eyes globose, separated by more than an eye width, slightly removed from anterior margin of pronotum; ocular setae present. Labium extending onto anterior region of metasternum; article I almost reaching posterior margin of bucculae; articles I and IV subequal in length, each longer than article II; article III about seven times as long as II (Fig. 1D). Antennae densely covered by golden pubescence and long brown setae; antennomere I thickest, curved laterally; II slightly thicker than III–IV; III–IV subequal in width; IV filiform (as in Fig. 1).

Thorax: Pronotum long, completely covering meso- and metanota; covered by fine pubescence intermixed with elongate brown setae; collar distinct, delineated by coarse punctures; anterior lobe with pair of small, frosty, pubescent areas formed by a very dense layer of short setae (Figs. 1C, 4A, 4B, 4D, 4E, 7C) (sometimes indistinct, as in Figs. 1A, 1B, 5A, 7A); humeri raised, prominent; posterior lobe with numerous punctures, without rounded or finger-like process at posterior margin. Forewing with four closed cells, brown, with white macula at basal cells adjacent to costal margin and at wing apex, without other marks centrally (Figs. 1A–1C, 4E, 7A, 7C); some brachypterous specimens lacking basal (Fig. 4C) or apical macula (Fig. 5A). Thoracic pleura with numerous suboval, coarse punctures (Figs. 5C, 7B, 7D, 7E). Meso- and metasterna each with a pair of tubercles; tubercles meeting at suture, forming a cavity between them. Metasternum with posterior margin convex; metasternal scent gland opening (=omphalium) obscure. Legs moderately stout, lacking prominent spines or teeth, light-brown at base, brown distally, with a faint yellowish annulus on femora; hind femur slightly incrassate; grasping comb occupying approximately half or less of male fore tibial length, absent in female; middle tibia with a distally decreasing row of elongate dark-brown trichobothria-like setae on distal third; all tarsi three segmented; tarsomere I shortest; fore leg with tarsomere III longer than II; middle and hind legs with tarsomere II longer than III; claws long, slender, slightly curved.

Abdomen: Numerous suboval, coarse punctures, except for laterotergites, sterna II–VI adjacent to lateral margins, and whole sternum VII (Figs. 2C, 2F) and segment VIII; prominent paired longitudinal carinae present along mediotergites II–III and basally on mediotergite IV (visible with wings open or removed) (Fig. 2C). Sides of abdominal sterna with narrow, roughly ovate (sometimes longitudinal) impressed furrows (=striae sensu Polhemus, 2021) (Figs. 7D, 7E). Male terminalia of moderate size; segment VIII without black denticles or projections; proctiger with a pair of anterodorsal projections (Figs. 3E–3H, 6H), lateral lobes without angular projections (Fig. 6F), black denticles absent; parameres symmetrical, slender, curved (Figs. 3E–3H, 6E). Female abdominal segment VIII on same plane as VII; first gonocoxae exposed, plate-like, black denticles absent; proctiger globose, button-like, longer than wide (Figs. 5C, 7B, 7D).

Etymology. The generic name Foveavelia is derived from fovea (Latin), meaning pit, referring to the coarse cuticular punctures present throughout the body, and Velia, the nominate genus of the family. Gender feminine.

Natural history. Three of the five species here included in Foveavelia are known by only one of the sexes, and two of them by just one specimen. Thus, specimens of these species are relatively rare in collections and not many details are known about its habitat and biology.

Discussion. After examining all described Veliinae from the Neotropical region to study the phylogenetic relationships of Paravelia, it was possible to define a set of features shared by only five species here included in the new genus Foveavelia: F. amapaensis (Rodrigues et al., 2014) n. comb., F. anta (Mazzucconi, 2000) n. comb., F. bilobata (Rodrigues et al., 2014) n. comb., F. dilatata (Polhemus & Polhemus, 1984) n. comb., and F. hungerfordi (Drake & Harris, 1933) n. comb. (see Table 1).

Table 1 Checklist of species of Foveavelia.

Taxon	Distribution	
Foveavelia amapaensis (Rodrigues et al., 2014)	Brazil (Amapá)	
Foveavelia anta (Mazzucconi, 2000)	Argentina (Salta), Paraguay (Concepción)	
Foveavelia bilobata (Rodrigues et al., 2014)	Brazil (Ceará, Mato Grosso), Colombia (Amazonas)	
Foveavelia dilatata (Polhemus & Polhemus, 1984)	Brazil (Amazonas, Pará), French Guiana (Saint-Georges), Guyana (Upper Demerara-Berbice), Peru (Madre de Dios), Suriname (Para)	
Foveavelia hungerfordi (Drake & Harris, 1933)	Brazil (Mato Grosso)	
Note:

All species are removed from the genus Paravelia.

Foveavelia is defined by the following combination of characteristics: (1) the unusual coarse cuticular punctures found throughout the thorax and abdomen (Fig. 2); (2) the pair of small, frosty pubescent areas formed by a very dense layer of short setae on anterior lobe of the pronotum (Figs. 1A, 4A, 4B, 4D, 4E); (3) the fore tibial grasping comb present only in males, occupying 1/6 (Fig. 1D) to 1/2 of the fore tibial length; (4) the middle tibia with a row of elongate dark-brown trichobothria-like setae on the distal third, decreasing in size distally; (5) the macropterous specimens with the apical macula of the forewings elongate and slightly constricted at mid-length, reaching the wing apex; (Figs. 1A–1C, 4E, 7A, 7C); (6) the meso- and metasterna each with a pair of tubercles, these tubercles meeting at suture, forming a cavity between them (Fig. 1D); (7) the metasternum with the posterior margin convex (Fig. 1D); (8) the male abdominal segment VIII acuminating posteroventrally (Figs. 2F, 3C, 3D, 6A, 6B); and (9) the male proctiger with a pair of anterodorsal projections (Figs. 3E–3H, 6H).

Rodrigues et al. (2014) and Rodrigues & Moreira (2016) had already indicated the similarities shared by these species, but did not propose any taxonomic changes. The main diagnostic feature of Foveavelia is the presence of coarse cuticular punctures along the body (Fig. 2). As described by Mazzucconi (2000), these structures are cuticular depressions with deep, transverse grooves, and a sensilla-like seta placed eccentrically to these grooves, or sometimes centrally. Although the arrangement of this seta varies, its size is similar among different punctures. Unlike the structure described above, the pronotal punctures typical of most Veliidae, including Foveavelia, are rounded and covered by a cluster of centrally directed microtrichia along the puncture rim, making the transverse grooves and sensilla-like seta difficult to see (Figs. 2A, 2B). On the other hand, the distinctive punctures of Foveavelia lack clusters of microtrichia along the puncture rim (Figs. 2D–2E). These punctures usually do not touch each other (Figs. 2C, 2F), although in some regions of the body, such as on the sides of abdominal mediotergites I–III, they are conjoined, forming larger, flower-like structures (Figs. 2A, 2D).

Because Paravelia is not monophyletic and due to the morphological heterogeneity among its species, it is difficult to detect diagnostic characters for the entire genus. Paravelia basalis (Spinola, 1837) (Fig. 8A), the type species, differs from Foveavelia in several characteristics, including the absence of coarse cuticular punctures throughout the body, the absence of frosty pubescence on the anterior lobe of the pronotum, the different shape and color of the forewing maculae, the presence of a pair of distinct projections on male abdominal sternum VII, the different shape of male abdominal segment VIII (Figs. 8B, 8C), and the presence of anterodorsal and anterolateral projections on the male proctiger (Fig. 8D).

Figure 8 Paravelia species.

(A–E) Paravelia basalis (Spinola, 1837), (A) dorsal habitus of male from Brazil, (B and C) male abdominal segment VIII, (B) lateral, and (C) dorsal views, (D) male genital capsule, white arrow indicates anterodorsal projection, black arrow indicates anterolateral projection, both on proctiger, (E) left paramere in anterolateral view. (F–G) Paravelia foveata Polhemus & Polhemus, 1984, (F) dorsal, and (G) ventral habitus of male from Brazil, dashed line indicates length of grasping comb.

Paravelia foveata Polhemus & Polhemus, 1984 (Figs. 8F, 8G) displays a pattern of cuticular punctures on the body similar to Foveavelia. However, because this species has several characteristics that are not present in other species here assigned to Foveavelia, it is not included in the new genus. The following features are exclusive to P. foveata: (1) general body color reddish-brown, with short pubescence; (2) antennomere IV very small, fusiform; (3) anterior lobe of pronotum with a pair of yellowish-white markings; (4) forewings with differently shaped closed cells, with an additional macula basally, and apical macula crescent-shaped, not reaching wing apex; (5) male fore tibial grasping comb occupying about two-thirds of the tibial length (Fig. 8G); (6) row of elongate dark-brown trichobothria-like setae on the middle tibia occupying half of the segment; (7) coarse punctures present on abdominal sternum VII; and (8) male proctiger without anterodorsal projections. The cuticular modifications found along the body of P. foveata, especially those present on the abdomen, are very likely not homologous to those displayed by Foveavelia. Cuticular modifications similar to those of Foveavelia, but probably also not homologous, are found in members of Microveliinae (Gerromorpha: Gerridae), such as Neoalardus typicus (Distant, 1903) and Hebrovelia singularis Lundblad, 1939.

Distribution. The species of the genus are distributed throughout South America east of the Andes, with published records from Argentina, Brazil, French Guiana, Guyana, Paraguay, Peru and Suriname (Fig. 9).

Figure 9 Geographic distribution records of Foveavelia species.

Images from Google Maps.

Foveavelia amapaensis (Rodrigues et al., 2014) n. comb.

(Figs. 1A, 3A, 3E, 7E, 9)

Paravelia amapaensis Rodrigues, Moreira, Nieser, Chen & Melo in Rodrigues et al., 2014: 5–6 (original description).

Diagnosis. Body length 4.80. Pronotum rhomboid. Distance between basal and apical forewing maculae greater than length of basal macula. Forewing basal macula of macropterous specimens elongate (Fig. 1A). Male abdominal sternum VII without pair of medial gibbosities; pair of posterior rounded lobes almost at level of posterior margin. Male proctiger with a pair of rounded-lobe projections anterodorsally. Paramere with laterodorsal margin slightly convex in lateral view (Fig. 3E).

Type locality. Brazil: Amapá: Santana, Porto Santana, I.C.O.M.I.

Repository. The male holotype was deposited at the Museu Nacional, Universidade Federal do Rio de Janeiro, Rio de Janeiro, Brazil. However, it was destroyed together with most of the entomological collection of the institution in the 2018 fire (Kury, Giupponi & Mendes, 2018).

Published records. Brazil: Amapá (Rodrigues et al., 2014).

Distribution. This species is known only from the type-locality (Fig. 9). The acronym I.C.O.M.I. refers to “Indústria e Comércio de Minérios S/A”, a mining company that was contracted by the government of the state of Amapá in the 1950s to build an ore loading dock at Porto Santana, on the estuary of the Amazon River (Bastos, Valente & Oliveira, 2021).

Discussion. This species was described based solely on the male holotype and some structures (e.g., tibial grasping comb, thoracic sterna, abdominal tergum VIII) were not examined, either because the specimen was glued to a paper card or because it was not possible to dissect the male abdominal segment VIII. The diagnostic features used here to separate it from congeners need to be better studied when more specimens become available, since variations are expected. In the original description, the authors neglected the pair of small lobes present near the posterior margin of male abdominal sternum VII (Fig. 3A), which are similar to those of P. bilobata (Fig. 3B).

Foveavelia anta (Mazzucconi, 2000) n. comb.

(Figs. 2, 3F, 9)

Paravelia anta Mazzucconi, 2000: 130–134 (original description).

Diagnosis [based on original description]. Body length 5.40–5.70. Pronotum rhomboid. Distance between basal and apical forewing maculae greater than length of basal macula. Forewing basal macula of macropterous specimens elongate. Male fore tibial grasping comb occupying 1/6 of the tibial length. Male abdominal sternum VII with a pair of medial gibbosities (see Mazzucconi, 2000, page 131, fig. 10); posterior margin without pair of rounded lobes. Male proctiger with a pair of distinct, divergent, spinose projections anterodorsally (Fig. 3F). Paramere with laterodorsal margin sinuous in lateral view, widened at middle (Fig. 3F).

Type locality. Argentina: Salta: Anta, 50 km East of Las Lajitas.

Repository. Museo Argentino de Ciências Naturales “Bernardino Rivadavia”, Buenos Aires, Argentina.

Published records. Argentina: Salta (Mazzucconi, 2000). Paraguay: Concepción (Mazzucconi, 2000).

Distribution. The two records of this species are located in the Río de La Plata basin, in southern South America (Fig. 9).

Foveavelia bilobata (Rodrigues et al., 2014) n. comb.

(Figs. 1B–1D, 3B–3D, 3G, 9)

Paravelia bilobata Rodrigues, Moreira, Nieser, Chen & Melo in Rodrigues et al., 2014: 8–10 (original description).

Diagnosis. Body length 5.03–5.07. Pronotum rhomboid. Distance between basal and apical forewing maculae greater than length of basal macula. Forewing basal macula of macropterous specimens elongate (Figs. 1B, 1C). Male fore tibial grasping comb occupying 1/4 of tibial length. Male abdominal sternum VII without pair of medial gibbosities; pair of posterior rounded lobes extended further than posterior margin (Fig. 3B). Male proctiger with a pair of bilobed projections anterodorsally. Paramere with laterodorsal margin sinuous in lateral view, widened at posterior half (Fig. 3G).

Type locality. Brazil: Mato Grosso: Nova Xavantina, Reserva Biológica Municipal Mário Viana (Parque Municipal do Bacaba), Córrego Bacaba.

Repository. DPIC.

Published records. Brazil: Ceará and Mato Grosso (Rodrigues et al., 2014; Rodrigues & Moreira, 2022).

Distribution. This species has been recorded from the Caatinga and Cerrado biomes in northeastern and central-western Brazil, respectively. Here, we extend its distribution considerably to the west, based on material from the Colombian Amazon (Fig. 9).

Discussion. We examined a macropterous female (Figs. 7C, 7D) from northeastern Brazil deposited in the NMNH that most likely belongs to this species. However, because the sample lacked a male for studying the terminalia and given that F. amapaensis and F. bilobata are very similar, we refrained from definitely assigning such female to the latter species.

Type material examined. HOLOTYPE: BRAZIL, Mato Grosso, Nova Xavantina, Parque Municipal do Bacaba, Córrego Bacaba, 14°43′14.80″S, 52°21′35.63″W, 11.X.2003, S.O. Pagioro col. (macropterous ♂, DPIC). PARATYPE: Mato Grosso, Córrego da Mata, quarta ordem, 15°01′32″S, 52°26′29″W, 17.XI.2005, H.S.R. Cabette et al. col. (1 macropterous ♂, MZUSP).

Additional material examined. COLOMBIA, Amaz. [=Amazonas], Leticia, III-12–15-1969, P. & P. Spangler (1 macropterous ♂, NMNH).

Foveavelia dilatata (Polhemus & Polhemus, 1984 ) n. comb.

(Figs. 3H, 4–6, 9)

Paravelia dilatata Polhemus & Polhemus, 1984: 498 (original description).

Diagnosis. Body length 5.35–6.50. Sexually dimorphic; pronotum subtriangular, distinctly widened anteriorly in the male (Figs. 4D, 4E), rhomboid in the female (Figs. 4A, 4B). Distance between basal and apical forewing maculae greater than length of basal macula. Forewing basal macula of macropterous specimens short, roughly ovate (Fig. 4E). Male fore tibial grasping comb occupying almost half of tibial length. Male abdominal sternum VII with a weak median carina extending along segment, without medial gibbosities or rounded lobes posteriorly. Male proctiger with a pair of spinose anterodorsal projections, directed laterally (Figs. 3H, 6H). Paramere with laterodorsal margin sinuous in lateral view, widened at middle (Fig. 6F).

Supplemental description. Male abdominal segment VIII with posterodorsal margin almost straight medially; posterolateral corners rounded, slightly extended posteriorly; posterolateral margin excavated; posteroventral margin narrowed centrally, rounded (Figs. 6A–6C).

Type locality. Suriname: Para: Coesewijneproject, 12 km West of Saramacca-brug.

Repository. Zoölogisch Museum, Rijksuniversiteit te Utrecht, Utrecht, The Netherlands.

Published records. Brazil: Amazonas, Pará (Polhemus & Polhemus, 1984; Pereira & Melo, 2007; Polhemus, 2014; Rodrigues et al., 2014; Rodrigues & Moreira, 2016; dos Santos et al., 2021). French Guiana: Saint-Georges (Armisén et al., 2022). Guyana: Upper Demerara-Berbice (Polhemus, 2014). Suriname: Para (Polhemus & Polhemus, 1984).

Distribution. This species is distributed from the Guianas to the southern portion of the Peruvian Amazon (Fig. 9).

Discussion. This species was described based on brachypterous specimens from Suriname and Brazil (Polhemus & Polhemus, 1984). Rodrigues et al. (2014) illustrated the macropterous male, and Rodrigues & Moreira (2016) illustrated the brachypterous female. The size and shape of the forewing maculae change according to the wing condition. In macropterous specimens, the basal and apical maculae are larger; the basal macula is roughly ovate and the apical macula is elongate and slightly constricted at mid-length (Fig. 4E). The maculae can be smaller and fainter in brachypterous specimens (Figs. 4A, 4B, 4D, 5A); the basal macula, when present, is oval, and the apical macula can be small and rounded (Figs. 4B, 4D) or display the typical shape seen in macropterous specimens (Fig. 4C).

Type material examined. PARATYPES: BRAZIL, Amazonas, Reserva Ducke, 25 km NNE Manaus, 120 m, July 21, 1973, R.T. Schuh, impounded area in forest stream, Paratype Paravelia dilatata J.T. & D.A. Polhemus, J.T. Polhemus Collection 2014 C.J. Drake Accession (1 brachypterous ♀, NMNH).

Additional material examined. BRAZIL, Amazonas: Igarape [=Igarapé] da Anta, Reserva Ducke, 25 km NE of Manaus, 60 m, 24.5 °C, 25 August 1989, CL2472, D.A. & J.T. Polhemus (4 brachypterous ♂, 3 brachypterous ♀, NMNH); Manaus, Reserva Florestal Adolpho Ducke, poça temporária na trilha para o igarapé Barro Branco, 02˚53′S, 59˚58′W, 27.XI.2012, U.G. Neiss col. (1 brachypterous ♀, MZUSP); Barcelos, Serra do Aracá, Igarapé Ataiana, tributário do Rio Negro, 00°88′56.57″S, 62°54′13.90″W, 10.VIII.2009, N. Hamada, R.L. Ferreira-Keppler, A.M.O. Pes & C.A.S. Azevêdo col. (1 macropterous ♂, INPA). PERU, Madre de Dios: unnamed stream at Finca Las Piedras, 27 July 2022, L-2099, Coll: R.W. Sites, 12°13′37″S, 69°7′01″W, 214 m, standing stream w/leafpacks & marginal vegetation (1 brachypterous ♀, UMC).

Foveavelia hungerfordi (Drake & Harris, 1933) n. comb.

(Figs. 7A, 7B, 9)

Velia hungerfordi Drake & Harris, 1933: 46 (original description).

Paravelia hungerfordi: J. Polhemus, 1976: 512 (changed combination).

Diagnosis. Body length 4.80. Pronotum rhomboid. Distance between basal and apical forewing maculae smaller than the length of the basal macula (Fig. 7A). Forewing basal macula of macropterous specimens elongate.

Type locality. In the original description, the authors mentioned only "Chapada, Brazil", without additional data. The locality very likely corresponds to Chapada do Guimarães, state of Mato Grosso, and the type material was probably collected by Herbert Huntington Smith (Moreira et al., 2011).

Repository. Carnegie Museum of Natural History, Pittsburgh, United States.

Published record. Brazil: Mato Grosso (Drake & Harris, 1933).

Distribution. Known only from the type-locality in central-western Brazil (Fig. 9).

Discussion. Drake & Harris (1933) described this species based on two female specimens. Because only the type series is known, the comparison with males of other species is not possible. Mazzucconi (2000) provided a redescription of this species and compared it with F. anta. She distinguished the two based mainly on the length and width of the body, the shape of the anterolateral margin of the pronotum, and the size of the forewing maculae. Females of Foveavelia are very similar and the condition of the forewing maculae is the only viable character to identify F. hungerfordi.

Type material examined. PARATYPE: BRAZIL, Mato Grosso: Chapada, Brazil, Acc. No. 2966, Paratype Velia hunferfordi D&H, CJ Drake Coll. 1956, Velia hunferfordi D&H. (1♀ macropterous, NMNH).

Key to the species of Foveavelia

Foveavelia amapaensis and F. bilobata are known only from the male, whereas F. hungerfordi is known only from the female. As in other Neotropical Veliinae genera, males should be prioritized for identification purposes, as they usually display the most informative diagnostic features. 1. Distance between basal and apical forewing maculae in macropterous specimens smaller than length of basal macula (Fig. 7A)F. hungerfordi

- Distance between basal and apical forewing maculae in macropterous specimens greater than length of basal macula (Figs. 1A–1C, 4E)2

2. Pronotum sexually dimorphic, subtriangular in male (Figs. 4D–4E), rhomboid in female (Figs. 4A–4B); male fore tibial grasping comb occupying almost half of tibial length; male proctiger with a pair of small, spinose, and laterally directed anterodorsal projections (Figs. 3H, 6H)F. dilatata

- Pronotum not sexually dimorphic, rhomboid in both sexes (Figs. 1A–1C); male fore tibial grasping comb occupying 1/6–1/4 of tibial length; male proctiger with pair of distinct and upward directed anterodorsal projections (Figs. 3E–3G)3

3. Male abdominal sternum VII with a pair of medial gibbosities, without rounded lobes at posterior margin; male proctiger with a pair of long, laterally divergent spines on anterodorsal region (Fig. 3F)F. anta

- Male abdominal sternum VII without pair of medial gibbosities, with a pair of rounded lobes at or near posterior margin (Figs. 3A, 3B); male proctiger with a pair of dorsally directed rounded single or bilobed projections on anterodorsal region (Figs. 3E, 3G)4

4. Pair of rounded lobes on male abdominal sternum VII almost at level of posterior margin (Fig. 3A); projections of male proctiger unilobed (Fig. 3E); laterodorsal margin of paramere slightly convex in lateral view (Fig. 3E)F. amapaensis

- Pair of rounded lobes on male abdominal sternum VII extended posteriorly further than posterior margin (Fig. 3B); projections of male proctiger bilobed (Fig. 3G); laterodorsal margin of paramere sinuous in lateral view (Fig. 3G)F. bilobata

Discussion

In terms of phylogenetic relationships, the only species of Foveavelia heretofore included in a study is F. dilatata (Armisén et al., 2022). It was recovered as sister to the genus Stridulivelia, which displays different types of cuticular modifications on the thorax and abdomen (glabrous longitudinal striae or elongate lacunae). In the future, expanding the taxonomic scope of Veliinae in a phylogenetic analysis would be interesting to test whether all taxa with cuticular modifications, including Stridulivelia, Foveavelia and P. foveata are closely related.

Conclusions

After examination of all american species within the subfamily Veliinae, a new Neotropical genus has been established to accommodate five species previously classified in Paravelia. This new genus has been characterized morphologically using SEMs and photographs. Future phylogenetic hypotheses are required to elucidate the closely related lineages of this new genus.

Supplemental Information

Supplemental Information 1 Foveavelia species, dorsal and ventral views.

(A) Foveavelia amapaensis (Rodrigues et al., 2014), dorsal habitus of male holotype (specimen was destroyed in a fire). (B–D) Foveavelia bilobata (Rodrigues et al., 2014), (B) dorsal habitus of male from Colombia, (C) dorsal, and (D) ventral habitus of male paratype from Brazil, dashed line indicates length of grasping comb. am = apical macula, bm = basal macula, fb = frosty pubescence.

Supplemental Information 2 Structures of Foveavelia anta (Mazzucconi, 2000). Scanning electron microscopy.

(A) Dorsal view of head, pronotum and part of abdomen (wings removed). (B) Pronotal punctures. (C) Male abdomen in dorsal view (wings and genital capsule removed). (D) Part of abdominal mediotergite II with suboval punctures. (E) Suboval puncture of abdominal mediotergite IV in detail. (F) Male abdomen in lateral view (wings and genital capsule removed), white arrow indicates posteroventral margin of abdominal segment VIII acuminating distally. mt = mediotergite, lt = laterotergite. Photographs provided by Dr. Silvia Mazzucconi.

Supplemental Information 3 Foveavelia species, male structures.

(A–B) Ventral view of abdominal apex, (A) F. amapaensis (Rodrigues et al., 2014), (B) F. bilobata (Rodrigues et al., 2014). (C–D) Abdominal segment VIII of F. bilobata, (C) lateral, and (D) ventral views. (E–H) Genital capsule in lateral view, showing proctiger projections detailed in frontal view, (E) F. amapaensis, (F) F. anta (Mazzucconi, 2000), (G) F. bilobata, (H) F. dilatata (Polhemus & Polhemus, 1984).

Supplemental Information 4 Foveavelia dilatata (Polhemus & Polhemus, 1984), dorsal views of female and male.

(A–B) Dorsal habitus of brachypterous females, (A) paratype from Brazil, (B) specimen from Brazil. (C) Dorsal view of abdomen of brachypterous female, specimen from Brazil. (D) Dorsal habitus of brachypterous male, paratype from Brazil. (E) Dorsal habitus of macropterous male, specimen from Brazil.

Supplemental Information 5 Foveavelia dilatata (Polhemus & Polhemus, 1984), brachypterous female from Peru.

(A) Dorsal, (B) ventral, and (C) lateral habitus.

Supplemental Information 6 Male structures of Foveavelia dilatata (Polhemus & Polhemus, 1984), paratype from Brazil.

(A–C) Abdominal segment VIII, (A) lateral, (B) ventral, and (C) dorsal views. (D) Genital capsule in lateral view. (E) Left paramere in lateral view. (F–H) proctiger, (F) dorsal, (G) lateral, and (H) frontal views, white arrow indicates pair of anterodorsal projections.

Supplemental Information 7 Foveavelia species, dorsal and lateral views.

(A–B) Foveavelia hungerfordi, macropterous female paratype from Brazil, (A) dorsal, and (B) lateral views. (C–D) Foveavelia sp., macropterous female from Brazil, (C) dorsal, and (D) lateral views. (E) Foveavelia amapaensis, macropterous male holotype in lateral view (specimen was destroyed in fire). if = impressed furrow.

Supplemental Information 8 Paravelia species.

(A–E) Paravelia basalis (Spinola, 1837), (A) dorsal habitus of male from Brazil, (B–C) male abdominal segment VIII, (B) lateral, and (C) dorsal views, (D) male genital capsule, white arrow indicates anterodorsal projection, black arrow indicates anterolateral projection, both in proctiger, (E) left paramere in anterolateral view. (F–G) Paravelia foveata Polhemus & Polhemus, 1984, (F) dorsal, and (G) ventral habitus of male from Brazil, dashed line indicates length of grasping comb.

We are grateful to Dr. Silvia Mazzucconi (Universidad de Buenos Aires, Buenos Aires, Argentina) for permission to use the scanning electron microscopy photographs of Foveavelia anta, and Dr. Robert Sites (UMC) for allowing the inclusion of the record of F. dilatata from Peru, as well as for providing some of the photos of this species. We also thank Dr. Alan L. de Melo (DPIC), Dr. Marcelo Duarte (MZUSP), Dr. Marcio Oliveira and Msc. Thiago Mahlmann (INPA), and Dr. Thomas Henry (USNM) for providing access to their institutional collections. Critical reviews were kindly provided by Dr. Dan Polhemus (Bishop Museum, Hawaii, USA) and Dr. Daniel Reynoso-Velasco (Instituto de Ecología, Veracruz, Mexico).

Additional Information and Declarations

Competing Interests

Author Contributions

Data Availability

New Species Registration

The authors declare that they have no competing interests.

Higor D. D. Rodrigues conceived and designed the experiments, performed the experiments, analyzed the data, prepared figures and/or tables, authored or reviewed drafts of the article, and approved the final draft.

Felipe F. F. Moreira conceived and designed the experiments, analyzed the data, authored or reviewed drafts of the article, and approved the final draft.

The following information was supplied regarding data availability:

The raw data is available in the figures.

The following information was supplied regarding the registration of a newly described species:

Publication LSID: urn:lsid:zoobank.org:pub:DAAB68B0-7AB5-4D92-AAE8-A9051CD9EC11.

Genus Foveavelia LSID: urn:lsid:zoobank.org:act:5B804C14-919F-438B-BED3-224DA8720BCD.

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
