# Peer review of "Foveavelia, a new South American genus of Veliinae (Hemiptera: Heteroptera: Veliidae)"

_PeerJ, doi:10.7717/peerj.16772_

## Round 0.1 · original submission · Minor Revisions

Dear Drs. Rodrigues and Moreira:

Thanks for submitting your manuscript to PeerJ. I have now received two independent reviews of your work, and as you will see, the reviewers raised some concerns about the research. Despite this, these reviewers are optimistic about your work and the potential impact it will have on research studying veliids. Thus, I encourage you to revise your manuscript, accordingly, considering all the concerns raised by both reviewers.

The concerns of the reviewers are relatively minor; this, this is a minor revision. However, there are many suggestions provided by the reviewers (both have provided marked up versions of your manuscript), which I am sure will greatly improve your manuscript once addressed.

Therefore, I am recommending that you revise your manuscript, accordingly, considering all the issues raised by the reviewers. Once these issues are addressed, we may move towards acceptance of your work.

Good luck with your revision,

-joe

·

Basic reporting

This is a manuscript for the most part well-written, although comments/changes have been made/suggested to improve the English language.
Structure is logic, although I noticed that the species diagnoses do not contrast the same features. Not considering unique features, these sections should contrast the state character of the "same" features. Also, most of what is included in the final Discussion section (not that for each species) is part of the Results. The authors need to find the proper place for this information. I have suggested to accommodate it in a Discussion section within the description of the genus.

Experimental design

no comment

Validity of the findings

no comment

Additional comments

This is a valuable manuscript with important information on the taxonomy of a poorly classified genus, Paravelia. The authors have examined all the available material in order to have strong support for what the propose in the MS. Also, they are well-informed on the topic, as they have worked describing many new species in the genus Paravelia.
Numerous changes have been suggested directly on the pdf version of the MS.

·

Basic reporting

This paper proposes a new genus, Foveavelia, to hold five species currently placed in the genus Paravelia. As the authors properly , the current set of species currently held in Paravelia probably does not constitute a monophyletic assemblage, and this paper is another in a series of works over the past several decades that have removed various monophyletic segregates from it.

Experimental design

N/A

Validity of the findings

The new genus established here seems to be based on a reasonable set of synapomorphic characters, for which detailed photographic illustrations are provided, and the case for separating it from Paravelia is presented in detail. Excellent photographic and line drawing illustrations are also presented for all of the species now included in the new Foveavelia, along with a distribution map. Given this comprehensive treatment, the content of the paper does not require any revision.

Additional comments

In regard to format, a certain number of corrections to text have been suggested on the accompanying electronic copy of the manuscript text, highlighted in yellow. Once these minor revisions are made, the paper should be ready for publication in PeerJ.

---

## Round 0.2 · accepted · Accept

Dear Drs. Rodrigues and Moreira:

Thanks for revising your manuscript based on the concerns raised by the reviewers. I now believe that your manuscript is suitable for publication. Congratulations! I look forward to seeing this work in print, and I anticipate it being an important resource for groups studying veliids. Thanks again for choosing PeerJ to publish such important work.

Best,

-joe